



**Uncertainties of ground-based microwave radiometer retrievals in**
**zenith and off-zenith methods under snow conditions**
Wengang Zhang[1], Guirong Xu[1,*], Yuanyuan Liu[2], Guopao Yan[3], Shengbo Wang[4]
[1] Hubei Key Laboratory for Heavy Rain Monitoring and Warning Research, Institute of Heavy Rain,
China Meteorological Administration, Wuhan, China
[2] Hubei Meteorological Information and Technological Support Center, Wuhan, China
[3] Wuhan Meteorological Bureau of Hubei Province, Wuhan, China
[4] Mewbourne School of Petroleum & Geological Engineering, University of Oklahoma, Norman,
USA
[*] Correspondence to: Guirong Xu (grxu@whihr.com.cn)
**Abstract.** This paper is to investigate the uncertainties of microwave radiometer
(MWR) retrievals in snow conditions and also explore the discrepancies of MWR
retrievals in zenith and off-zenith methods. The MWR retrievals were averaged in the
±15 min period centered at sounding times of 00:00 and 12:00 UTC and compared
with the radiosonde observations (RAOBs). In general, the MWR retrievals have a
better correlation with RAOB profiles in off-zenith method than in zenith method,
and the biases (MWR observations minus RAOBs) and root mean square errors
(RMSEs) between MWR and RAOB are also clearly reduced in off-zenith method.
The biases of temperature, relative humidity, and vapor density decrease from 4.6 K,
9 %, and 1.43 g m$^{-3}$ in zenith method to -0.6 K, -2 %, and 0.10 g m$^{-3}$ in off-zenith



method, respectively. The discrepancies between the MWR retrievals and the RAOB
profiles along with the altitude present the same situation. Case studies show that the
impact of snow on accuracies of the MWR retrievals is more serious in heavy
snowfall than that in light snowfall, but the off-zenith method can mitigate the impact
of snowfall. The MWR measurements become less accurate in snowfall is mainly due
to the retrieving method which does not consider the effect of snow, and the
accumulated snow on the top of radome increases the signal noise of MWR
measurement. As the snowfall drops away by gravity in the sides of the radome and
the off-zenith observations are more representative of the atmospheric conditions for
RAOBs.
**Key words:** Microwave radiometer, Retrieval uncertainties, Off-zenith method,
Snowfall
**1. Introduction**
Atmospheric profiles of temperature, relative humidity, and vapor density can be
retrieved from ground-based microwave radiometer (MWR) measurements (Sánchez
et al. 2013; Ware et al. 2013). These profiles are available nearly continuously and are
extensively utilized in the forecasting and analysis of intense convective weather, also
they have been assimilated into numerical weather prediction models (Marzano et al.
2005; Knupp et al. 2009; Löhnert et al. 2012; Madhulatha et al. 2013). The instability
indices calculated from the MWR-retrieved thermodynamic atmospheric profiles are





also employed in operational meteorology (Chan et al. 2010; Cimini et al. 2015;
Leena et al. 2015). However, since the radiative transfer model used in the MWR
does not consider the impact of precipitation on the MWR brightness temperature
measurements, the MWR retrievals become less accurate under precipitation
conditions (Ware et al. 2004; Xu et al. 2014). To improve the accuracy of MWR
retrievals in rainy conditions, some methods are performed to minimize the influence
of liquid water on MWR measurements. The MWR is equipped with a hydrophobic
radome and a special blower system, which can sweep water beads and snow away
from the radome (Chan 2009). A method based on linear regression is also employed
to reduce the discrepancy between the MWR retrievals and the radiosonde
observation (RAOB) profiles (Sánchez et al. 2013). Recently, the off-zenith method is
applied in MWR observations and off-zenith retrievals provide higher accuracy
during precipitation by minimizing the effect of liquid water on the radiometer
radome (Cimini et al. 2011, 2015; Ware et al. 2013; Xu et al. 2014).

15       Snow, a special type of precipitation, has distinct scattering characteristics in the

microwave. Some methods are explored to investigate these characteristics and
discuss their utilization on the snow measurements (Matrosov et al. 2008; Löhnert et
al. 2011; Xie et al. 2012). The scattering signal of snow is highly dependent on the
assumption of snow shape and snow size distribution (SSD), especially for
large-sized parameters (Kneifel et al. 2010). Some studies have demonstrated that
snowfall can significantly reduce the measurement accuracy of MWR (Knupp et al.
2009; Cimini et al. 2011; Ware et al. 2013). However, few studies are reported on the





improvements of MWR measurement accuracies in snow conditions. Moreover, in contrast with rain, snow usually freezes on the top of the radome, and it is not easily blown away from the radome by the blower system attached on the MWR. Since MWR retrieval accuracies generally are better in off-zenith method than in zenith method under precipitation conditions (Xu et al., 2014) and snow does not easily accumulate on the sides of a radome, we attempt to employ off-zenith method to improve the MWR retrieval accuracies during snowfall.

This paper is organized as follows: Section 2 will briefly describe the data and methodology employed in this study; Section 3 compares the MWR-retrieved atmospheric profiles of temperature, relative humidity and vapor density with RAOB profiles obtained at Wuhan station, then discusses the accuracies of MWR retrievals under snow conditions and the effect of off-zenith method on it; and Section 4 gives some conclusions.

## 2. DATA AND METHODOLOGY

The data used in this study are collected in the Wuhan operational station (30.6 ° N, 114.1 ° E, and 23 m above sea level), including RAOB data, meteorological observation data and MWR data. The distances between them are all less than 30 m. RAOB data is the operational data, which is obtained at 00:00 and 12:00 UTC every day. The profiles of temperature and relative humidity are obtained by the Chinese GTS1-2 digital radiosonde at a high vertical resolution of 10 m, and the profiles of vapor density can be calculated from them. The meteorological observation data are



used to confirm the snowfall cases. The MWR data used in this paper is provided by
a MP-3000A unit manufactured by Radiometrics, observing at 2 elevation angles
(zenith and 15 ° elevation) up to 10 km. The MWR data has a higher temporal
resolution of ~3 min, and the vertical intervals are 50 m from the surface to 500 m,
100 m to 2 km, and 250 m to 10 km (Ware et al. 2013; Xu et al. 2014).

6        The MP-3000A unit observes brightness temperature at up to 35 channels,

including 21 K-band (22－30 GHz) and 14 V-band (51－59 GHz). Moreover, an
infrared radiation thermometer (IRT) is equipped on the MWR, which measures sky
infrared temperature at one zenith infrared (9.6–10.5 μm) channel and gives
information on cloud-base temperature (Ware et al. 2013; Cimini et al. 2015; Xu et al.
2015). Meteorological sensors attached to the MWR can obtain ambient temperature,
pressure, and relative humidity at the instrument level. The retrieved algorithm
developed by the factory can automatically convert the microwave, infrared, and
surface meteorological measurements into temperature, humidity, and liquid profiles
using radiative transfer equations with the aid of neural networks (Xu et al. 2015).
The neural network retrieval method uses historical radiosondes to characterize states
of the atmosphere that commonly occur at a particular location (Ware et al. 2013). A
five-year data set of historical radiosondes in Wuhan was used for neural network
training (Xu et al. 2014).

20       Three snow cases (shown in Table 1) are selected to present the comparison of the

profiles between MWR and RAOB under snow conditions, and the effect of
off-zenith method on improving the MWR measurement accuracy during snowfall is





explored. All cases in this study include at least one RAOB profile during snowfall.
Since it takes 30 minutes for the balloon from the surface to 10 km altitude in
sounding, the MWR retrievals were averaged in the ±15 minute period centered at
sounding times of 00:00 and 12:00 UTC and compared with the RAOB profiles.
Considering the vertical resolution of the RAOB profiles is not consistent with that of
MWR retrievals, the RAOB profiles are interpolated to the height levels of the MWR
retrievals. Based on the above process, there are eight temporal pairs of MWR and
RAOB profiles for comparison in this study. Methods used in this study are simply
employed to calculate the correlation coefficients, bias (MWR observation minus
RAOB), and root mean square error (RMSE) between the MWR and the RAOB for
each parameter in zenith and off-zenith methods. The discrepancies between MWR
retrievals and RAOB profiles at different heights are also calculated to explore how
the MWR retrievals accuracies vary with height.
**3. RESULTS ANALYSIS**
**3.1 Uncertainties of MWR retrievals in zenith and off-zenith methods under**
**snow conditions**

18       To explore the effect of off-zenith method on MWR measurement accuracy, the

simultaneous MWR zenith and off-zenith retrievals around the time of 00:00 and
12:00 UTC are compared with the RAOB profiles. Table 2 presents the comparison
of MWR retrievals against RAOB profiles in zenith and off-zenith methods under
snow conditions without considering the level division in altitude. All the MWR



retrievals have a better correlation in off-zenith method than that in zenith method
especially for relative humidity, and the biases and RMSEs are also clearly reduced in
off-zenith method. For temperature, the MWR zenith observations have a warm bias
of 4.6 K against RAOBs while in off-zenith method the bias decrease to -0.6 K, with
RMSE also decreasing from 5.7 K to 2.0 K. The MWR-retrieved relative humidity
has poor agreement with RAOB relative humidity in zenith method but reasonable in
off-zenith method, and the bias and RMSE also decrease from 10 % and 33% in
zenith method to -2 % and 20 % in off-zenith method, respectively. For vapor density,
the correlation coefficient between MWR observations and RAOBs increases from
0.7130 in zenith method to 0.9389 in off-zenith method. In zenith method, the bias is
1.43 g m$^{-3}$ with a RMSE of 2.14 g m$^{-3}$, while in off-zenith method both of them
decrease to 0.10 g m$^{-3}$ and 0.66 g m$^{-3}$, respectively. Obviously, the MWR retrievals
have better accuracies against RAOBs in off-zenith method than in zenith method.

14        To further compare the uncertainties of MWR retrievals against RAOBs in

zenith and off-zenith methods, the discrepancies between the MWR retrievals and the
RAOB profiles along with the altitude under snow conditions are also investigated.
As shown in Fig. 1, the temperature correlation coefficients in zenith method are
smaller than those in off-zenith method below 6 km especially around 3.75 km where
the correlation coefficient rapidly increases from 0.01 to 0.92, but the situation is
opposite above 6 km. The MWR temperature shows a warm bias against RAOB in
zenith method and the bias is larger than 3 K at most heights, while in off-zenith the
bias becomes cold and within -1 K at most heights. Both the MWR temperature





RMSEs in zenith and off-zenith methods approximately increase with height, but the
RMSE is clear smaller in off-zenith method. The MWR temperature RMSE is greater
than 4 K above 0.5 km in zenith method while in off-zenith method it is within 2 K at
most heights.

5        Fig. 2 presents the results for the relative humidity profiles. The correlation

coefficients between MWR observations and RAOBs are negative at most heights
below 2.5 km. Compared with zenith observations, off-zenith observations have well
agreement with RAOBs above 4.5 km. The correlation coefficient cannot be
calculated in some altitudes because the compared RAOB relative humidity remains
constant at these altitudes, so some breakpoints are shown in the Fig. 2a. The biases
of zenith and off-zenith observations are negative below 5 km and there are no
distinct differences between them. Above 6 km, both the biases in zenith and
off-zenith methods increase with height, but the bias is clear smaller in off-zenith
method. It is the same situation for the RMSE, the RMSE differences between zenith
and off-zenith observations are not evident below 5 km, while above 5 km the RMSE
is clearly smaller in off-zenith observations.

17       The comparison results for the vapor density profiles are shown in Fig. 3. It can

be seen that the correlation coefficient in zenith observation is positive below 3.5 km
but mostly negative above 3.5 km, while in off-zenith observation it is positive except
around 3 km. In general, the correlation coefficient is more reasonable in off-zenith
method than in zenith method. The bias of vapor density in zenith observation
increases from 0 g m$^{-3}$ at surface to 5.51 g m$^{-3}$ at 2 km and then decreases to near 0 g



$m^{-3}$ at 10 km again, but in off-zenith observation the bias is clear smaller with a value
within $\pm 1.0$ g $m^{-3}$. Both the RMSEs in zenith and off-zenith observations vary
similarly with height, in which the RMSE in zenith (off-zenith) observation firstly
increases to 3 km (2.3 km) and then decreases to near 0 g $m^{-3}$ at 10 km. Although the
RMSE has a close value in zenith and off-zenith observations, it is also clear smaller
in off-zenith observation. The RMSE in zenith observation is mostly greater than 1.0
g $m^{-3}$ with a peak of 2.60 g $m^{-3}$, yet it is generally smaller than 1.0 g $m^{-3}$ with a peak
of 1.47 g $m^{-3}$.

9        Based on the above analysis, it is clearly that snowfall has a significant impact

on MWR measurement accuracy, and off-zenith method can improve the accuracies
of MWR retrievals under snow conditions, especially for the temperature and vapor
density retrievals. Snowfall, one of precipitation, does not be considered in the MWR
retrieving method, so the MWR-retrieved atmospheric profiles in snow conditions are
not reasonable as those in non-precipitation conditions (Xu et al, 2014). Although a
special blower system is used to sweep water beads and snow away from the radome,
snowfall, particularly heavy snowfall will always freeze on the radome in the low
temperature situation. Snow produces a strong scattering signal in the microwave
region and the snow ice will increase signal noise of MWR measurement, so the
frozen snow on the radome will have great influence on the MWR measurement of
brightness temperature. Compared to zenith method, off-zenith method has better
measurement accuracies under snow conditions. This is mainly because that the
MWR observes at 15 ° elevation through vertical sections of the inverted "U" shaped



radome that are more readily cleared of snow/water droplets by gravity than the
horizontal sections observed at zenith. Moreover, the MWR accuracies are related
with the balloon drifting in sounding due to the wind in atmosphere (Xu et al., 2015),
and the off-zenith observations are more representative of the conditions in which
radiosonde observations are also taken (Xu et al., 2014), thus the MWR measurement
accuracies are generally better in off-zenith method than in zenith method.
**3.2 Case study**
To better understand the effect of off-zenith method on the improvement of
MWR retrieval accuracy, the comparison between the time series of the MWR
retrievals in a heavy snowfall and a light snowfall are performed. The heavy snowfall
happens from 00:07 UTC 5 February to 04:15 UTC 7 February in 2014 with
cumulative snowfall of 28.0 mm and the light snowfall happens from 07:16 UTC 8
February to 04:22 UTC 9 February in 2014 with cumulative snowfall of 2.3 mm.
As shown in Fig. 4, the MWR-retrieved temperature in zenith method presents a
clear increase at ~2.5 km in the heavy snowfall, but the increase is not clear in
off-zenith method. The MWR-retrieved temperature in zenith method is about 10 K
warmer than that in off-zenith method when the snowfall happens, and the greater
temperature is well accordant with the snowfall time. The clearly warmer temperature
disappeared in 1 h after the end of heavy snowfall. Fig. 5 illustrates the situation in
the light snowfall. The MWR temperature discrepancies between zenith and
off-zenith methods are not significant as those in the heavy snowfall, and the MWR
temperatures in zenith method are about 3 K warmer than those in off-zenith method





at ~2.5 km when the snowfall happens. The greater temperature is not obvious when

light snowfall maybe due to the light snow on the radome is blown away immediately

by the special blower system.

The MWR-retrieved temperatures have well agreement between zenith and

off-zenith method in light snow condition, while they have poor agreement in heavy

snowfall. Although a special blower system is used to sweep water beads and snow

away from the radome, the heavy snowfall is hardly blown away and will easily froze

on the radome. Frozen snow will have great influence on the MWR measurement of

brightness temperature, so heavy snowfall has more effect on the MWR observations

comparing with light snowfall. The greater temperature in zenith method is probably

caused by the discrepancies of MWR-measured brightness temperature, and this will

helpful to explain why the greater temperature is significant in heavy snow condition.

Off-zenith method significantly minimizes contamination from ice and snow, so the

MWR-retrieved temperature in zenith method is more reasonable especially when

heavy snowfall.

The MWR relative humidity discrepancies in zenith and off-zenith methods are

also significant in the heavy snowfall (Fig. 6). Although the MWR relative humidity

presents good agreement in zenith and off-zenith methods below 2.5 km, the MWR

relative humidity retrievals in zenith method are clear larger than those in off-zenith

method above 5 km, about 40 % at 7 km. Greater MWR relative humidity appears

above 7 km in zenith method and also well consistent with the timing of the heavy

snowfall, while this situation disappears in the off-zenith method. However, the



discrepancies between zenith and off-zenith methods are not clear in the light

snowfall, and the variation of the relative humidity is also more stable (Fig. 7). The

bottom of atmosphere is almost saturated when snowfall happens and we attribute

that snow will easily sublimation in the blowing of the special thermodynamic blower

system.

The situation for the vapor density is the same as the temperature. As shown in

Fig. 8, the MWR vapor density retrievals in zenith method are significantly larger

than those in off-zenith method at ~2.5 km in the heavy snowfall, and the time of

vapor density increasing is also consistent with the heavy snowfall time. The heavy

snowfall will also reduce the retrieval accuracies of vapor density by influencing the

brightness temperature measurements of MWR, thus the trend of vapor density

variation in zenith method is similar to that of temperature with heavy snowfall.

While in off-zenith method, the MWR vapor density retrievals are more reasonable

without the significantly larger area. In the light snowfall (Fig. 9), the MWR vapor

density retrievals present a similar trend in zenith and off-zenith methods, but the

former is clearly larger than the latter below 3 km.

Obviously, the MWR retrieval discrepancies between zenith and off-zenith

methods are greater in heavy snowfall than that in light snowfall, As mentioned

before, this is mainly because that the snowfall is more easy to freeze on the radome

top in heavy snowfalls and the signal noise caused by snowfall increases, while in the

sides of the radome the snowfall drops to the ground for gravity, so the MWR

retrieval discrepancies are greater in heavy snowfalls. However, in light snowfalls,





the blower system can sweep some snowfall away, so the impact of snowfall is not

greater as that in heavy snowfalls.

## 4. CONCLUSIONS

In this paper, the MWR retrieval accuracies in snow conditions are discussed by

comparing with the RAOBs and improvements of off-zenith method are also

investigated when snowfall happens. We also present two snowfall cases to explore

the MWR retrieval accuracy in heavy and light snow conditions. Based on the above

analysis, we draw the following conclusions:

1. Without considering the division of altitude, all the MWR retrievals have a

better correlation with RAOB profiles in off-zenith method than that in zenith method

especially for relative humidity when snowfall happens, and the biases and RMSEs

are also clearly reduced in off-zenith method. The temperature bias and RMSE

decrease from 4.6 K and 5.7 K in zenith method to -0.6 K and 2.0 K in off-zenith

method, respectively. The relative humidity bias and RMSE also decrease from 10%

and 33% in zenith method to -2 % and 20 % in off-zenith method, respectively, while

the correlation coefficient increases from 0.2531 to 0.7997. For vapor density, the

bias is 1.43 g m$^{-3}$ with a RMSE of 2.14 g m$^{-3}$ in zenith method, while in off-zenith

method the bias decreases to 0.10 g m$^{-3}$ with a smaller RMSE of 0.66 g m$^{-3}$.

2. The discrepancies between the MWR retrievals and the RAOB profiles along

with the altitude under snow conditions are also investigated. The MWR temperature



shows a warm bias against RAOB in zenith method and the bias is larger than 3 K at

most heights, while in off-zenith the bias becomes cold and is within -1 K at most

heights. The temperature RMSE is greater than 4 K above 0.5 km in zenith method

while in off-zenith method it is within 2 K at most heights. The vapor density

retrievals show the same situation, the bias and RMSE are clear smaller in off-zenith

method than in zenith method at most height. The off-zenith relative humidity

retrievals show a better agreement with RAOBs above 4.5 km but the correlation

coefficients are negative in zenith method. Although the differences between zenith

and off-zenith methods in relative humidity bias and RMSE are insignificant below 5

km, the bias and RMSE are clearly smaller in off-zenith method above 6 km.

3. Case studies show that the heavy snowfall has an obvious impact on the

accuracies of MWR retrievals by influencing the MWR brightness temperature

measurements, and the off-zenith method greatly mitigates the impact of snowfall.

The zenith retrievals have an increase trend during heavy snowfall process, but the

MWR retrievals in off-zenith method are smooth without the higher retrievals

appearing in zenith method.

4. The MWR measurements become less accurate in snowfall is mainly due to

the retrieving method which does not consider the effect of snow, moreover, the

snowfall accumulating on the radome especially in heavy snowfalls also increases the

signal noise of MWR measurement. As the snowfall drops away by gravity in the

sides of the radome and the off-zenith observations are more representative of the

atmospheric conditions for RAOB, the off-zenith method makes a positive effect on





mitigating the impact of snowfall,

## Acknowledgements

This study was supported by the Hubei Meteorological Bureau project under Grant
2014Q03, the open project of Institute of Plateau Meteorology, CMA, Chengdu under
Grant LPM2014009 and the National High Technology Research and Development
Program ("863" Program) of China under Grant 2012AA120902.

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

**Tables:**
Table 1. Details of three snow cases used in this study

| Start time of snowfall | End time of snowfall | Cumulated snowfall (mm) |
| --- | --- | --- |
| 00:07 UTC 5 Feb 2014 | 04:15 UTC 7 Feb 2014 | 28.0 |
| 07:16 UTC 8 Feb 2014 | 04:22 UTC 9 Feb 2014 | 2.3 |
| 12:00 UTC 17 Feb 2014 | 01:38 UTC 18 Feb 2014 | 11.1 |

Table 2. Comparison of MWR retrievals against RABOs in zenith and off-zenith
methods under snow conditions when not considering the level division in altitude.



| Parameters | Observation mode | Number of samples | Correlation coefficient | Bias | RMSE |
|---|---|---|---|---|---|
| Temperature | Zenith | 464 | 0.9239 | 4.6 K | 5.7 K |
| | Off-zenith | 464 | 0.9890 | -0.6 K | 2.0 K |
| Relative Humidity | Zenith | 464 | 0.2531 | 8.9 % | 33.1 % |
| | Off-zenith | 464 | 0.7997 | -2.2 % | 20.2 % |
| Vapor density | Zenith | 464 | 0.7130 | 1.43 g m$^{-3}$ | 2.14 g m$^{-3}$ |
| | Off-zenith | 464 | 0.9389 | 0.10 g m$^{-3}$ | 0.66 g m$^{-3}$ |

3 **Figures:**





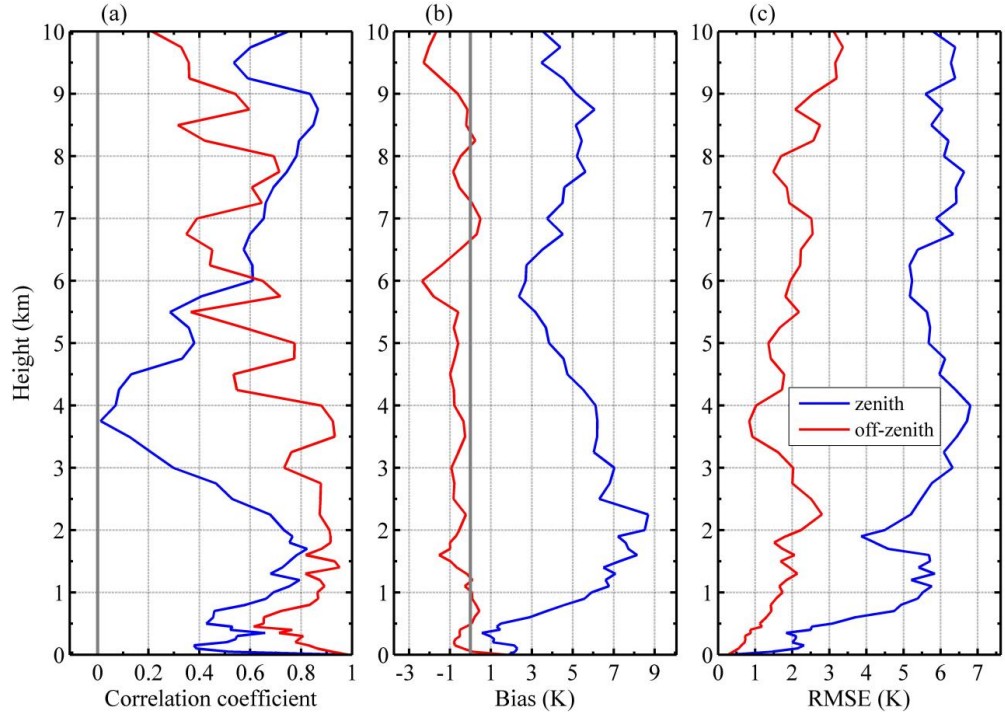

2    Figure 1. The correlation coefficient (a), bias (b) and RMSE (c) between the MWR

3    and RAOB temperature in zenith (blue) and off-zenith (red) observations.





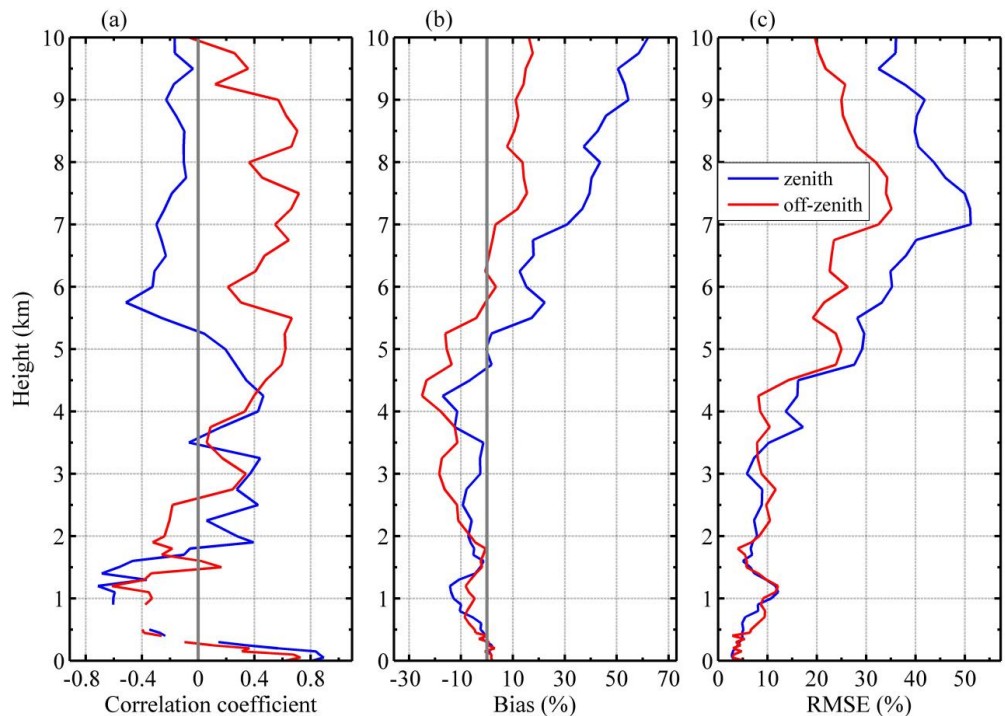

Figure 2. Same as Fig. 1 but for relative humidity profiles. Some breakpoints are

shown in Fig. 2a because the compared RAOB relative humidity remains constant at

these altitudes.





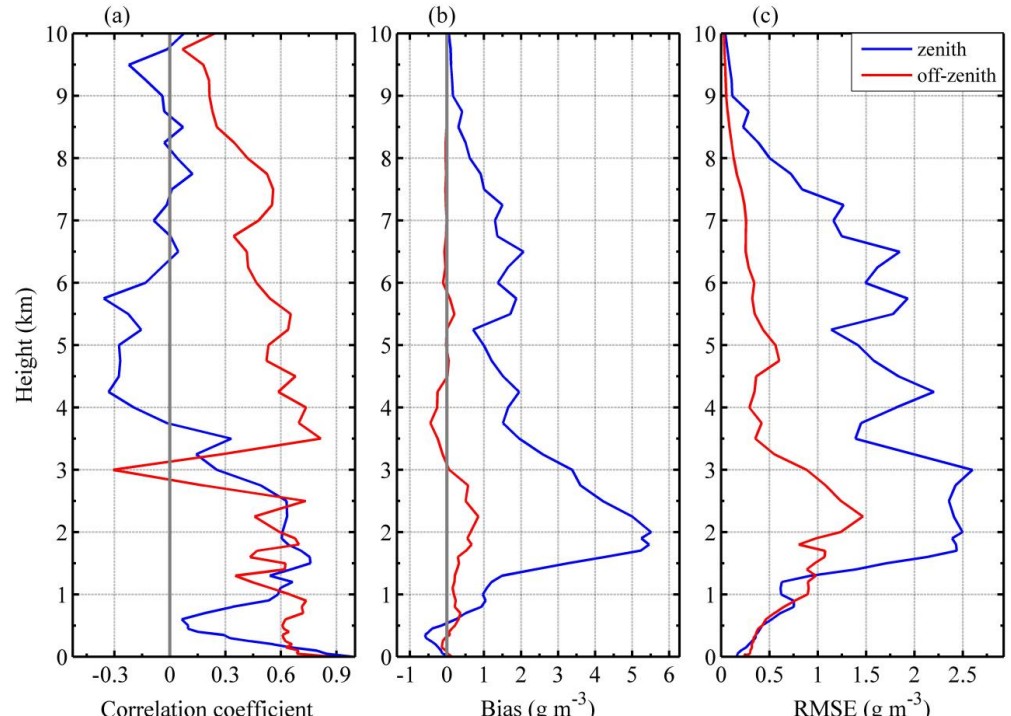

2    Figure 3. Same as Fig. 1 but for vapor density profiles.



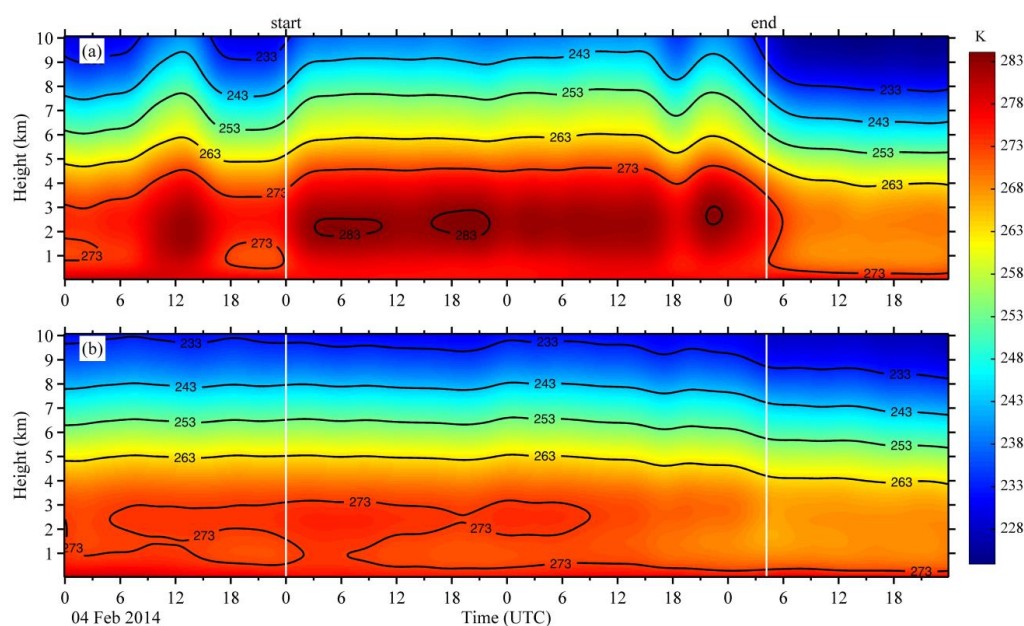

2 Figure 4 Comparison of temperature retrievals between zenith (a) and off-zenith (b)

3 observation in heavy snow condition. The start and end times of snowfall are

4 indicated by the vertical lines. The time series starts at 00:00 UTC 04 Feb 2014.



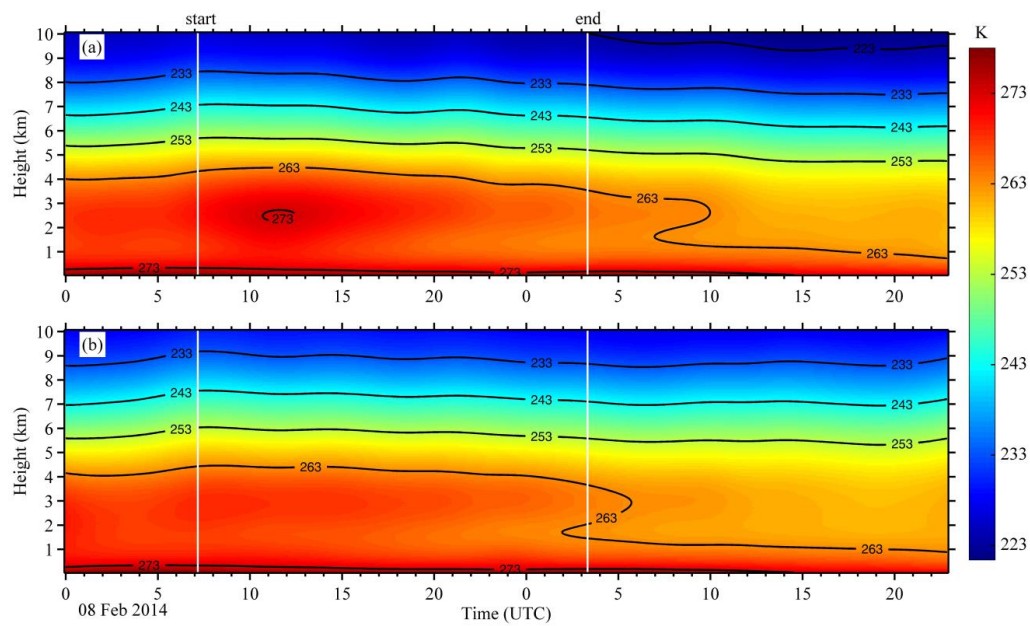

2    Figure 5 Comparison of temperature retrievals between zenith (a) and off-zenith (b)

3    observation in light snow condition. The start and end times of snowfall are indicated

4    by the vertical lines. The time series starts at 00:00 UTC 08 Feb 2014.



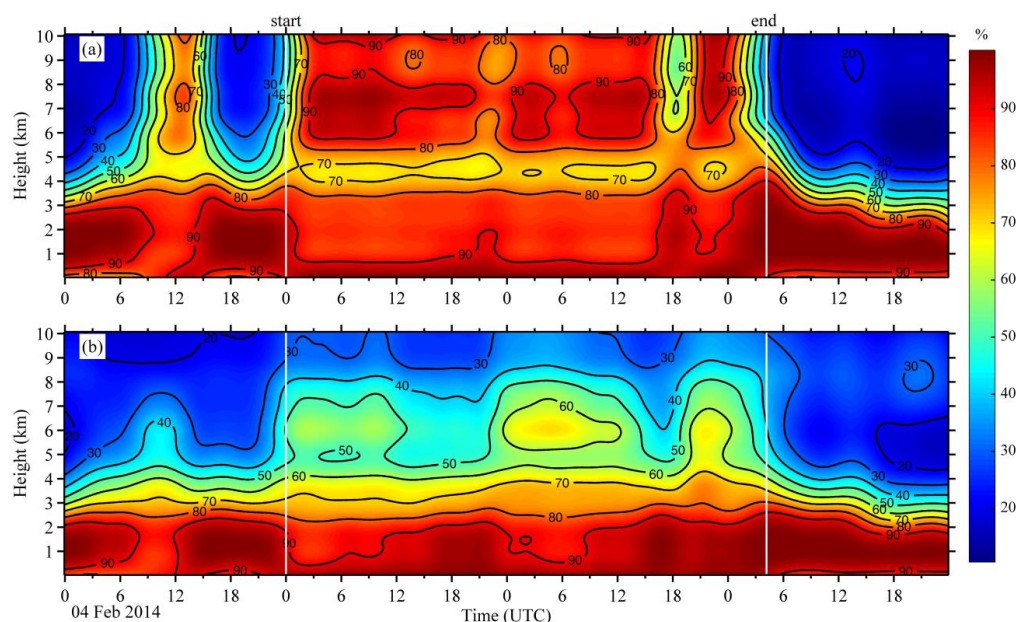

Figure 6 Same as Fig. 4 but for relative humidity retrievals.

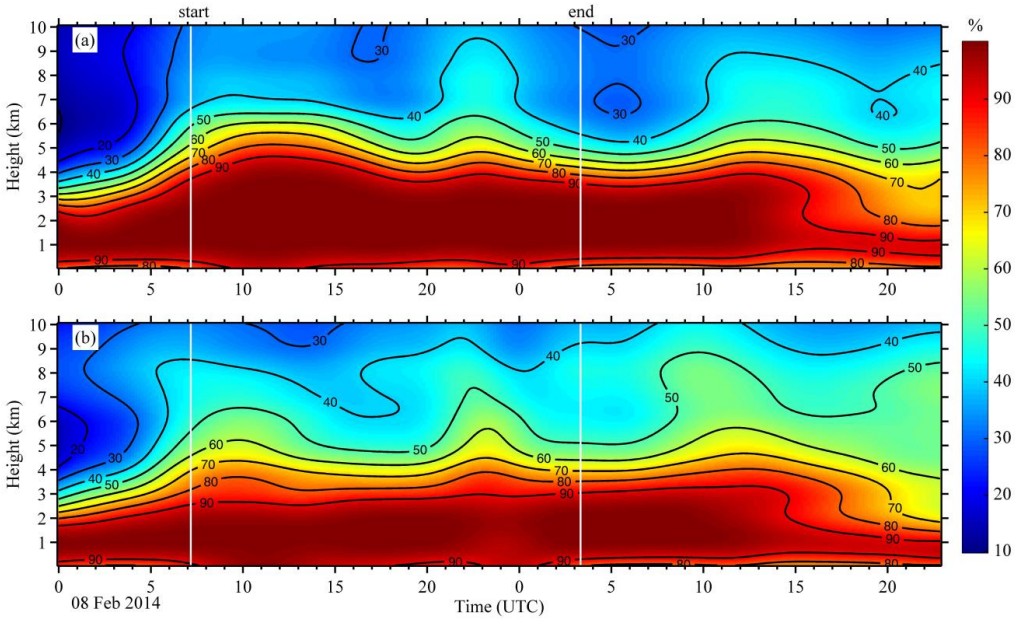





1    Figure 7 Same as Fig. 5 but for relative humidity retrievals.

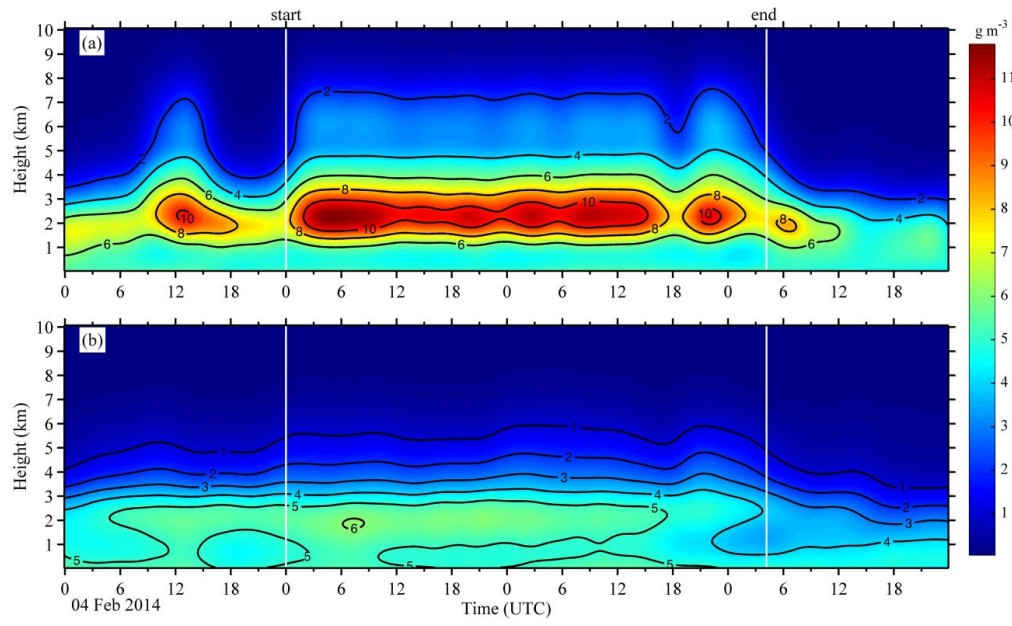

4    Figure 8 Same as Fig. 4 but for vapor density retrievals.





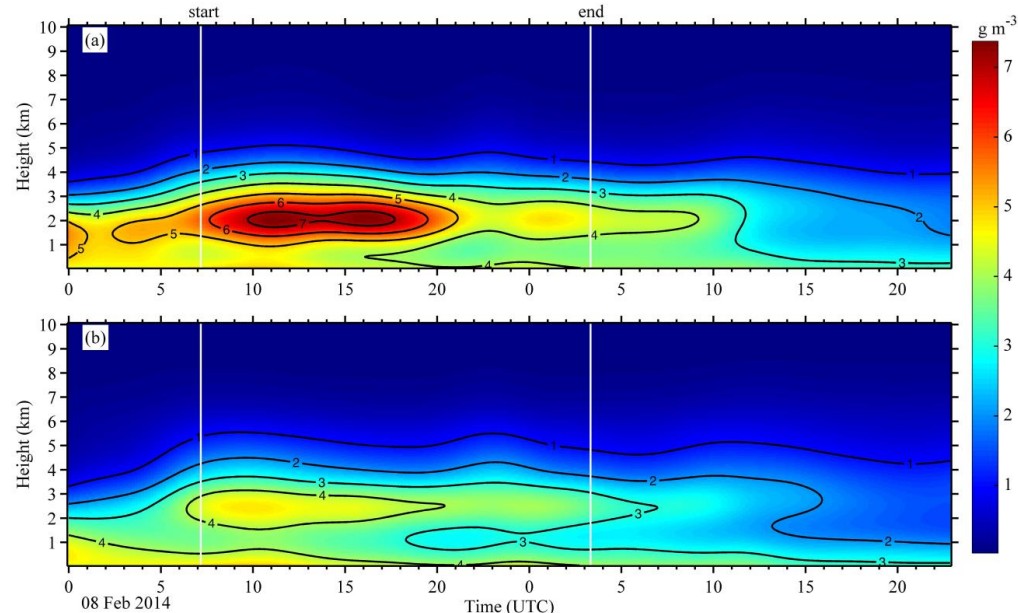

2  Figure 9 Same as Fig. 5 but for vapor density retrievals.