# Peer review of "Uncertainties of ground-based microwave radiometer retrievals in zenith and off-zenith methods under snow conditions"

_Atmospheric Measurement Techniques, 2016_

## Referee Comment (RC1) · Anonymous Referee #1 · 15 Sep 2016

This paper investigates the accuracy of microwave retrievals under snow conditions. With this goal the authors assess the accuracy of temperature, relative humidity and vapor density from microwave retrievals comparing with the profiles obtained from an in-situ technique as it is the radiosonde measurements. Microwave profiles are obtained from what authors call "two methods", that what it really means if the use of a neural network method for two different observations, zenith and off-zenith observations (with an elevation angle of  $15^{\circ}$ ). The authors compare the retrievals from both methods with the RS measurements for three snow cases. In addition, they compare the results obtaining from both microwave retrievals for two different events (heavy and light snow conditions).

The topic by itself is interesting, since microwave measurements have shown good results under clear and even cloudy conditions but the results are more problematic under rain and snow. However, I do not see any relevant result in this study that can be considered as an improvement of microwave technique under snow conditions or can help for a better understanding of the influence of snow on microwave measurements.

Next, I indicate the main reasons for my decision:

What the authors present as an improved method it is just the use of the neural network method for one observational angle (off-zenith, 15° elevation angle). At any moment there was something new in the method, as for example it could has been the consideration of snow properties (scattering, snow size distribution, etc.) in the microwave retrievals. So it cannot be considered as a new method or an improved method. It is well known that elevation scanning measurements increase the accuracy of retrieved microwave profiles (Crewell and Löhnert, IEEE 2007). So the fact that they use for the second retrieval a different observational angle to the zenith is not new at all.

The only explanation that the authors give about why one retrieval (off-zenith retrieval) give better agreement with RSs than the other one (zenith retrieval) it that there is more ice or snow in the part of the window where the mirror point for the zenith observation, and that due to the shape of the window (inverted "U") there is less snow for low elevation angles because the snow fall down more easily due to the gravity effect. From this explanation, I do not learn anything about how snow should be treated in microwave retrievals to improve the results. The only that I can learn is that I should clean the window. I miss in this study an evaluation of the effect of the snow on microwave measurements due to atmospheric emission.

The authors assess the uncertainties of the microwave retrievals only for three snow events. If I understand correctly, they compare microwave profiles with three radiosondes. This is totally insufficient to obtain any significant result of the uncertainties of microwave retrievals under these conditions.
The discussion about the two case studies is very speculative. The authors compare the temperature, relative humidity and vapor density profiles for both microwave retrievals given as the best results the ones from the off-zenith methods because it looks more reasonable. However, they should compare with independent measurements in order to confirm that what they consider reasonable is the real state of the atmosphere.

I do not give more specific comments about the study because I consider that in the current status this study should not be accepted in Atmospheric Measurement Technique.

**Reference**

Crewell, S. and Lohnert, U.: Accuracy of boundary layer temperature profiles retrieved with multifrequency multiangle microwave radiometry, Geoscience and Remote Sensing, IEEE Transactions on, 45, 2195–2201, 2007.

**AMTD**

---

## Referee Comment (RC2) · Anonymous Referee #2 · 20 Sep 2016

The manuscript AMT-2016-253 by Zhang et al. is within the scope of the journal and it meets the scientific quality for AMT.

However, before the manuscript gets accepted for publication on AMT, I encourage the authors to address the following comments that should help improving the overall quality of the manuscript.

Major Comment:

The manuscript it's not totally convincing unless the authors also show zenith and off-zenith retrievals under non-precipitating conditions, so to demonstrate that there's no systematic issue with zenith retrievals. Figures 4-9 seem to qualitatively suggest that

zenith and off-zenith retrievals are closer during non-precipitating conditions, but a statistical analysis, similar to Figures 1-3 but in non-precipitating conditions, would demonstrate that quantitatively. I strongly suggest the authors to add this analysis. It could be condensed in one figure with 3 panels showing RMSE for temperature (panel 1), relative humidity (panel 2), and vapour density (panel 3), each with zenith and off-zenith method in non-precipitating conditions.

Minor Comments:

- Several typos are present: e.g. page 2 (line 5), page 3 (line 12), page 5 (line 12 retrieved -> retrieval), ... Many times "clear" and "clearly" are misused: e.g. page 9 (lines 1, 5, and 9) But I stop here and leave these to the technical editor.

- page 4, line 18: "The distances between them are all less than 30 m." Please rephrase to clarify that the distances between MWR, RAOB launching station, and meteorological sensors are all less than 30 m. Observations may be much more distant due to radiosonde drifting, among other reasons.

- page 5, line 3: "up to 10 km" Please remove "up to 10 km" as it is incorrect and does not add anything here. 10 km is just the upper boundary of the vertical range for which the MWR software compute retrievals. Technically speaking the penetration depth depends upon absorption, i.e. it's different for each MWR channel.

- page 5, line 15: "radiative transfer equations" Please rephrase to clarify that radiative transfer model is used in the training phase of the retrieval algorithm, not in the real-time retrieval computation.

- page 6, line 6: "the RAOB profiles are interpolated to the height levels of the MWR" Interpolation does not account for the inherent MWR smoothing error. Ideally one should smooth the RAOB profiles at the original resolution considering the MWR averaging kernels and then interpolate on the MWR levels. E.g. see: http://www.atmos-meas-tech.net/5/1121/2012/ http://www.atmos-meas-tech.net/7/3023/2014/ The authors shall at least mention this issue.

- page 7, lines 18-19: "where the correlation coefficient rapidly increases from 0.01 to 0.92" The above sentence is misleading; it seems to hint that the correlation coefficient increases in a continuous way from 0.01 to 0.92, while it's either 0.01 (zenith) or 0.92 (off-zenith). I suggest to remove it.

- page 9, lines 7-8: "yet it is generally smaller than" I believe this refers to off-zenith, but this information is missing.

- page 9, lines 13-14: "are not reasonable as those" I believe the authors mean "are not as reasonable as those". Please check.

- page 9, line 19: "great" I suggest replacing this word with "some", as otherwise the authors should say with respect to what (similarly on page 11, line 8).

- page 10, lines 2-6: "the off-zenith observations are more representative of the conditions in which radiosonde observations are also taken" It's not clear whether the paper Xu et al. 2014 analyses data from the same site and synoptical conditions. If so, please state that clearly. Otherwise I believe their results cannot be generalised to the site/conditions presented in the manuscript. (similarly on page 14, lines 21-22)

- page 10, lines 17-18: "the greater temperature is well accordant with the snowfall time" In Figure 4 I see the warming of zenith retrievals during the snowfall. But I also see a warmer spot before the snowfall (around 12 UTC of 4 Feb). This is also evident in relative humidity and vapour density retrievals (Fig. 6 and 8, respectively). The authors completely ignore this feature, while I believe it must be discussed. Maybe there was liquid precipitation? A time series of precipitation rate and type would be very useful.

- page 11, lines 1-3: It seems to me obvious that the less snow, the less impact; so it is reasonable that heavy snow causes 10 K contrast, while light snow causes 3 K contrast. I don't see why the authors say that "light snow on the radome is blown away immediately"? The effect is there, 3 K it's far from being negligible.

- page 11, lines 10-12: The authors shall dwell more on the reason why snow causes larger temperature and humidity retrievals. I think Kneifel et al. 2010 provide some qualitative explanation.

- page 11, line 14: "temperature in zenith method is more reasonable" I believe the author mean in off-zenith

- page 12, lines 1-5: Not clear, please check grammar and possibly rephrase.

- page 13, lines 1-2: Please check grammar and possibly remove. I think it is obvious that larger impact is associated to heavier snowfall.

---

## Author Comment (AC1) · 7 Nov 2016

**Referee #1**

**Comment 1**

**Q:** Microwave profiles are obtained from what authors call "two methods", that what it really means if the use of aneural network method for two different observations, zenith and off-zenith observations (with an elevation angle of 15∘).

**A:** "Two methods" in this manuscript is actually means "two observation methods". We have replaced "methods" with "observations" in the manuscript for better understanding.

**Comment 2**

Q: However, I do not see any relevant result in this study that can be considered as an improvement of microwave technique under snow conditions or can help for a better understanding of the influence of snow on microwave measurements.

A: As well known, the measurement accuracy of microwave radiometer under precipitation condition is not good as that under non-precipitation, and currently there is no method that can completely eliminate the impact of precipitation especially for retrieval algorithm method. However, the off-zenith observation suggested by Radiometrics Corporation is proved to have positvie effect on reducing the impact of rainfall (Cimini et al., 2011; Ware et al., 2013; Xu et al., 2014). And this paper shows the snowfall also impacts the MWR measurement accuracy, but the impact of snowfall can partly reduce in the off-zenith observation. We added some discussion about the influence of snow on microwave measurements in the manuscript (Page 14, lines 6-21). As the snowfall cases are few, studies on how snow types and amount affect the measurement of MWR are not performed. This work can be carried out when we collect reasonal samples of snowfall.

**Comment 3**

Q: What the authors present as an improved method it is just the use of the neural network method for one observational angle (off-zenith, 15∘ elevation angle). At any moment there was something new in the method, as for example it could has been the consideration of snow properties (scattering, snow size distribution, etc.) in the microwave retrievals. So it cannot be considered as a new method or an improved method. It is well known that elevation scanning measurements increase the accuracy of retrieved microwave profiles (Crewell and Löhnert, IEEE 2007). So the fact that they use for these cond retrieval a different observational angle to the zenith is not new at all.

A: As mentioned above, there is no method that can completely eliminate the impact of precipitation/snow on MWR measurement accuracy especially for retrieval algorithm method by now. The off-zenith observation was suggested several years ago

and has positive effect on reducing the impact of precipitation. The purpose of this paper is to investigate the discrepancy of MWR retrievals under zenith and off-zenith observations, and check whether the off-zenith observation also has the positive effect in snowfall conditions as it plays in rainfall conditions.

Q: The only explanation that the authors give about why one retrieval (off-zenith retrieval)give better agreement with RSs than the other one (zenith retrieval) is that there is more ice or snow in the part of the window where the mirror point for the zenith observation,and that due to the shape of the window (inverted "U") there is less snow for low elevation angles because the snow fall down more easily due to the gravity effect. From this explanation, I do not learn anything about how snow should be treated in microwave retrievals to improve the results. The only that I can learn is that I should clean the window. I miss in this study an evaluation of the effect of the snow on microwave measurements due to atmospheric emission.

A: The ice or snow on the radome has significant effect on the measurements of MWR, and cleaning the window when snowfall happens will be very useful to decrease the impact of snow or ice. However, in cases that there is no one in situ the MWR site all day under snowfall conditions, the off-zenith observation can be a better way to reducing the impact of snowfall on MWR measument accuracy. Although the off-zenith observation can not completely eliminate the impact of snowfall, the MWR retrievals in off-zenith observation are closer to the RAOB profiles than those in zenith observation especially when the snow/ice on the window is not cleaned in time. We added some discussion about the influence of snow on microwave measurements in the manuscript (Page 14, lines 6-21). As the snowfall cases are few, studies on how snow types and amount affect the measurement of MWR are not performed. This work can be carried out when we collect reasonable samples of snowfall.

Q: The authors assess the uncertainties of the microwave retrievals only for three snow events. If I understand correctly, they compare microwave profiles with three radiosondes. This is totally insufficient to obtain any significant result of the uncertainties of microwave retrievals under these conditions.

A: As few snowfall happens in Wuhan each year, only three snow events are used in this study, but there are eight temporal pairs of MWR and RAOB profiles (page 6, lines 21-22). According to the analysis in this study, the authors think the conclusions are reasonable. Of course, sufficient snowfall cases can benefit our better understanding on the impact of snowfall, such as snow types and amount, and we will carry out this work when enough snowfall cases available.

Q: The discussion about the two case studies is very speculative. The authors comparethe temperature, relative humidity and vapor density profiles for both microwave retrievals given as the best results the ones from the off-zenith methods

because it looks more reasonable. However, they should compare with independent measurements in order to confirm that what they consider reasonable is the real state of the atmosphere.

A: The results on Figs. 2-4 present that the MWR retrievals in off-zenith observation are closer to the RAOB profiles than those in zenith observation, which indicate the MWR retrievals in off-zenith observation is more reasonable. The case study shows the detail look on the MWR measurement discrepancy under zenith and off-zenith observations during the whole snowfall process, for confirming again that the MWR retrievals in off-zenith observation is more reasonable. In this section, we add the precipitation rate and type observed by a disdrometer in the same site (Fig. 5, page11, lines 12-21). This may be helpful for the discussion on snow cases study.

---

## Author Comment (AC2) · 7 Nov 2016

**Referee #2**

**Major Comment**

**Q:**The manuscript it's not totally convincing unless the authors also show zenith and off-zenith retrievals under non-precipitating conditions, so to demonstrate that there's nosystematic issue with zenith retrievals. Figures 4-9 seem to qualitatively suggest thatzenith and off-zenith retrievals are closer during non-precipitating conditions, but a statistical analysis, similar to Figures 1-3 but in non-precipitating conditions, would demonstrate that quantitatively. I strongly suggest the authors to add this analysis. It could becondensed in one figure with 3 panels showing RMSE for temperature (panel 1), relative humidity (panel 2), and vapor density (panel 3), each with zenith and off-zenithmethod in non-precipitating conditions.

**A:**The comparison between zenith and off-zenith observations in non-precipitating conditions was studied by Xu et.al (2014). We add the citation of results from this paper and also showed the RMSE of temperature, relative humidity and vapor density under non-precipitating conditions, using the data around the time of snowfall.The results are shown in Fig.1and page 8 (lines10-14).

Minor Comments

**Q:** Several typos are present: e.g. page 2 (line 5), page 3 (line 12), page 5 (line 12retrieved -> retrieval),Many times "clear" and "clearly" are misused: e.g. page 9(lines 1, 5, and 9) But I stop here and leave these to the technical editor.

**A:**"retrieving method" is replaced by "retrievalalgorithm".
"off-zenith retrievals" means "profiles retrieved from MWR off-zenith observations", so we think the formulation is ok.
"Retrieved algorithm" is replaced by "retrieval algorithm".
The misused "clear" and "clearly"have been rectified.

Q: page 4, line 18: "The distances between them are all less than 30 m." Please rephraseto clarify that the distances between MWR, RAOB launching station and meteorological sensors are all less than 30 m. Observations may be much more distant due toradiosonde drifting, among other reasons.

A: we rephrase to "The distances between RAOB launching station, disdrometer, MWR and meteorological sensors are all less than 30 m, but the distance between sounding profile and MWR retrieval in high altitude may become larger due to radiosonde drifting." (Page 4, lines 20-22)

Q: page 5, line 3: "up to 10 km" Please remove "up to 10 km" as it is incorrect anddoes not add anything here. 10 km is just the upper boundary of the vertical

rangefor which the MWR software computes retrievals. Technically speaking the penetrationdepth depends upon absorption, i.e. it's different for each MWR channel.

A: "up to 10 km"is removed.

Q: page 5, line 15: "radiative transfer equations" Please rephrase to clarify that radiativetransfer model is used in the training phase of the retrieval algorithm, not in the real-time retrieval computation.

A: We rephrase the citation. "The retrieval algorithm developed by the factory can automatically convert the microwave, infrared, and surface meteorological measurements into temperature, humidity, and liquid profiles with the aid of neural networks (Xu et al. 2015).Long time radiosondes and liquid water content profiles that generated from radiosondes were proceed within a radiative transfer model and will be used as the neural network training set (Ware et al. 2013)." (Page 6, lines 2-8)

Q: page 6, line 6: "the RAOB profiles are interpolated to the height levels of the MWR"Interpolation does not account for the inherent MWR smoothing error. Ideally oneshould smooth the RAOB profiles at the original resolution considering the MWR averaging kernels and then interpolate on the MWR levels.The authors shall at least mention this issue.

A: we used this method in this paper and table 2、Fig2-4 are reworked according to the new data. (Page 6, lines 19-21; page 20 table 2; pages 22-24 Fig 2-4)

Q: page 7, lines 18-19: "where the correlation coefficient rapidly increases from 0.01 to0.92" The above sentence is misleading; it seems to hint that the correlation coefficientincreases in a continuous way from 0.01 to 0.92, while it's either 0.01 (zenith) or 0.92(off-zenith). I suggest to remove it.

A: We improve the description to avoid misleading. "As shown in Fig.2, the temperature correlation coefficients in zenith observation are smaller than those in off-zenith observation below 6 km, but the situation is opposite above 6 km." (Page 8, lines 15-17)

Q: page 9, lines 7-8: "yet it is generally smaller than" I believe this refers to off-zenith, but this information is missing.

A: "yet it is generally smaller than 1.0 g m$^{-3}$ with a peak of 1.47 g m$^{-3}$ in off-zenith observation." (Page 10, lines 4-5)

Q: page 9, lines 13-14: "are not reasonable as those" I believe the authors mean "are

not as reasonable as those". Please check.

A: "Snowfall, as one of precipitations, does not be considered in the MWR retrieval algorithm, so the MWR-retrieved atmospheric profiles in snow conditions are not as reasonable as those in non-precipitation conditions." (Page 10, lines 9-11)

Q: page 9, line 19: "great" I suggest replacing this word with "some", as otherwise theauthors should say with respect to what (similarly on page 11, line 8).

A: "great" is replaced by "some".

Q: page 10, lines 2-6: "the off-zenith observations are more representative of the conditions in which radiosonde observations are also taken" It's not clear whether the paperXu et al. 2014 analyses data from the same site and synopticalconditions. If so, please state that clearly. Otherwise I believe their results cannot be generalised to thesite/conditions presented in the manuscript. (Similarly on page 14, lines 21-22)

A: the paper Xu et al. 2014 analyses data from the same site but the conditions is not the same as this manuscript. Xu discussed the precipitation condition but wefocus on snow condition. In this section, we try to discuss the probable reasons why off-zenith observation has well measurement accuracy. The radiosondes are drifting, so the RAOB profiles are different from MWR zenith retrievals. Off-zenith observation is slant and it is more similar to the radiosonde.

Q: page 10, lines 17-18: "the greater temperature is well accordant with the snowfalltime" in Figure 4 I see the warming of zenith retrievals during the snowfall. But I alsosee a warmer spot before the snowfall (around 12 UTC of 4 Feb). This is also evident inrelative humidity and vapour density retrievals (Fig. 6 and 8, respectively). The authorscompletely ignore this feature, while I believe it must be discussed. Maybe there wasliquid precipitation? A time series of precipitation rate and type would be very useful.

A: Yes, there was liquid precipitation at that time.We add the precipitation rate and type using the data collected by a disdrometer in the same site.(Fig. 5, page11, lines 12-21)

Q: page 11, lines 1-3: It seems to me obvious that the less snow, the less impact; soit is reasonable that heavy snow causes 10 K contrast, while light snow causes 3 Kcontrast. I don't see why the authors say that "light snow on the radome is blown awayimmediately"? The effect is there, 3 K it's far from being negligible.

A: "light snow on the radome will be blown away more easily". The $3\,\mathrm{K}$ in light snowfall may not be cased only by snowfall. But for heavy snowfall, 10 K is not reasonable. (Page 12, lines 12-14)

Q: page 11, lines 10-12: The authors shall dwell more on the reason why snow causeslarger temperature and humidity retrievals. I think Kneifel et al. 2010 provide somequalitative explanation.

A: we add the discussion about how snow causes larger temperature and humidity retrievals with the help of this paper (Kneifel et al. 2010). (Page 14, lines 6-21)

Q: page 11, line 14: "temperature in zenith method is more reasonable" I believe theauthor mean in off-zenith

A: Off-zenith observation significantly minimizes contamination from ice and snow, so the MWR-retrieved temperature in off-zenith observation is more reasonable especially when heavy snowfall. (Page 13, lines 2-4)

Q: page 12, lines 1-5: Not clear, please check grammar and possibly rephrase.

A: we rephrase "However, in light snowfall condition, the discrepancies of relative humidity between zenith and off-zenith observations are not clear and the variation with time is also more stable without the high relative humidity above 6 km that appeared in heavy snowfall condition (Fig. 9). " (page 13, lines 12-15)

Q: page 13, lines 1-2: Please check grammar and possibly remove. I think it is obviousthat larger impact is associated to heavier snowfall.

A: It is removed.

---

## Author Comment (AC3) · 7 Nov 2016

Thanks for all the referees to provide very useful comments and suggestions. We have improved the manuscript according to the suggestions and also point the page and line number where the modification positing in the revised manuscript. Please see our responses below.

Changes in the manuscript by us:
1. In the process of manuscript revising, Dejun Li provides us with disdrometer data and also helps us to discussing the measurements of disdrometer, so we add him as a co-author.
2. The time in Fig. 7,9,11 is mistaken, we correct it.

Below is response for comments:

**Referee #1**

Responses are shown in reply to comment1.

**Referee #2**

Responses are shown in reply to comment2.